# Beyond the Clinic: The Activation of Diverse Cellular and Humoral Factors Shapes the Immunological Status of Patients with Active Tuberculosis

**DOI:** 10.3390/ijms24055033

**Published:** 2023-03-06

**Authors:** Nancy Liliana Tateosian, María Paula Morelli, Joaquín Miguel Pellegrini, Verónica Edith García

**Affiliations:** 1Departamento de Química Biológica, Facultad de Ciencias Exactas y Naturales, Universidad de Buenos Aires, Intendente Güiraldes 2160, Pabellón II, 4°piso, Ciudad Universitaria, Buenos Aires C1428EGA, Argentina; 2Instituto de Química Biológica de la Facultad de Ciencias Exactas y Naturales (IQUIBICEN), Facultad de Ciencias Exactas y Naturales, Universidad de Buenos Aires, Consejo Nacional de Investigaciones Científicas y Técnicas, Intendente Güiraldes 2160, Pabellón II, 4°piso, Ciudad Universitaria, Buenos Aires C1428EGA, Argentina; 3Centre d’Immunologie de Marseille Luminy, INSERM, CNRS, Aix-Marseille Université, Parc Scientifique et Technologique de Luminy, Case 906, CEDEX 09, 13288 Marseille, France

**Keywords:** tuberculosis, infection, immunology, endotype

## Abstract

*Mycobacterium tuberculosis* (*Mtb*), the etiologic agent of tuberculosis (TB), has killed nearly one billion people in the last two centuries. Nowadays, TB remains a major global health problem, ranking among the thirteen leading causes of death worldwide. Human TB infection spans different levels of stages: incipient, subclinical, latent and active TB, all of them with varying symptoms, microbiological characteristics, immune responses and pathologies profiles. After infection, *Mtb* interacts with diverse cells of both innate and adaptive immune compartments, playing a crucial role in the modulation and development of the pathology. Underlying TB clinical manifestations, individual immunological profiles can be identified in patients with active TB according to the strength of their immune responses to Mtb infection, defining diverse endotypes. Those different endotypes are regulated by a complex interaction of the patient’s cellular metabolism, genetic background, epigenetics, and gene transcriptional regulation. Here, we review immunological categorizations of TB patients based on the activation of different cellular populations (both myeloid and lymphocytic subsets) and humoral mediators (such as cytokines and lipid mediators). The analysis of the participating factors that operate during active *Mtb* infection shaping the immunological status or immune endotypes of TB patients could contribute to the development of Host Directed Therapy.

## 1. Introduction

Every year, 10 million people fall ill with tuberculosis (TB). Although TB is a preventable and curable disease, 1.5 million people die from TB each year, making *Mycobacterium tuberculosis* (*Mtb*) the second leading infectious killer after the coronavirus SARS-CoV-2, the agent of the COVID-19 pandemic [1]. Human TB infection encompasses a continuous spectrum of stages. After initial exposure, *Mtb* may be eliminated by the host immune response, persist as a latent infection, or progress to primary active disease [2]. Then, TB disease may remain in a latent form, naturally progressing in a slow or quick manner to active TB or rounding through incipient and subclinical conditions before developing into symptomatic active disease or ultimate disease resolution. 

Even though the spectra of TB infection and disease stays continuous, categorizing *Mtb* into these discrete states may allow us to improve the knowledge of the pathogenesis of TB and the development of new diagnostic and therapeutic tools to prevent the progression to active TB disease and the ongoing transmission of the *Mtb* bacilli [2]. Accordingly, Imperial et al. identified TB phenotypes that would allow to define the duration of the duration of anti-TB treatment as an alternative to the ‘one-size-fits-all’ treatment currently used worldwide. In fact, the authors identified populations that were eligible for a 4-month treatment and characterized the phenotypes that are hard to treat and analyzed the impact of adherence and dosing strategy on outcomes. Patients with sputum smears < 2+ or without cavitation were cured with an antibiotic treatment spanning 4 months. In contrast, the presence of cavitations or elevated smears in TB patients required at least 6 months of anti-TB chemotherapy. They concluded that regimen extent can be chosen to improve outcomes, providing a stratified medicine method as an alternative to the universal treatment employed at present. 

New investigations are currently analyzing if certain TB clinical phenotypes could successfully be treated with shorter antimicrobial regimens [3]. For the last 50 years, the treatment of TB patients has been performed according to standardized approaches that do not consider the immune response, the variation in human susceptibility to infection, the pharmacokinetics of drugs, and the required individual duration of the treatment. “Precision Medicine”, in contrast, improves patient’s outcome because it involves prevention and treatment strategies that consider individual variability, predicting who will benefit most from specific therapies and who will not. In this context, TB immune endotypes are defined as the distinct molecular pathways through which an individual can progress to active TB, and might be classified as immune deficient or immune exuberant [4]. In this way, the evaluation of TB immune endotypes might contribute to the identification of individuals at risk for relapse versus patients that might be successfully treated with shorter regimens.

## 2. Clinical and Immunological Characterization of Patients with Active TB

The clinical presentation of active TB is variable among individuals. Actually, TB patients display differences in the severity of the disease, tissue pathology and bacillary burden [5]. The current guidelines for the diagnosis of pulmonary TB are based on the demonstration of acid-fast bacilli (AFB) on sputum microscopy and chest radiograph [6], although some countries have already incorporated the molecular test GeneXpert MTB/RIF. According to the radiological lesions, TB patients may be clinically classified as those with the “mild” phenotype or patients with a single lobe involved and without visible cavities; those with the “moderate” phenotype or patients presenting unilateral involvement of two or more lobes with cavities, if present, reaching a total diameter no greater than 4 cm; or those with “severe” phenotype, which corresponds to TB patients displaying bilateral disease, with massive affectation and multiple cavities (advanced disease) [7,8]. Furthermore, AFB in the sputum smear (bacillary burden) also helps with the clinical classification of TB patients [9]. Moreover, clinical symptoms analyzed in TB patients to establish the time of disease evolution include weight loss, night sweats, symptoms of malaise or weakness, persistent fever, presence of cough, history of shortness of breath and hemoptysis [7]. 

The immune defense against *Mtb* requires a balanced and efficient response from multiple cell types to generate an appropriate cell-mediated immunity. In particular, impaired immunity in TB infection is associated with a depressed Th1 cytokine response [10,11]; thus, IFN-γ secretion by peripheral blood mononuclear cells stimulated with sonicate *Mtb* is lower in patients with the most severe manifestation of TB [12]. Moreover, Th1 cells have been demonstrated to have a crucial function role in granuloma formation and clearance of *Mtb* [10,12]. IFN-γ, a cytokine produced primarily by T cells and NK cells, is an important mediator of macrophage activation in controlling several intracellular pathogens, including *Mycobacteria* [10]. Accordingly, it was proposed that exogenous IFN-γ might benefit IFN-γ-deficient patients but be detrimental to individuals producing too much IFN-γ [13]. Actually, deficiencies in the IL-12/IFN-γ-STAT1 signaling pathway led to the dissemination of mycobacterial infections [14,15]. IFN-γ, acting through the IFN-γR, has been shown to be an integral part of various antibacterial procedures such as granuloma formation and phagosome-lysosome fusion. As a result, its absence is associated with an overgrowth of intracellular pathogens and disease caused by *Mtb* [16]. 

Therefore, though the clinical phenotype is the visible manifestation of the disease, it represents the result of different functional, immunological and pathobiological mechanisms happening in each patient. Then, subjacent to clinical manifestations, the “immune endotype” in particular reflects the individual immunological and molecular mechanisms used by the patient to fight the pathogen [17], fluctuating with multiple different molecular pathways and pathologies. In the following sections, we review studies that describe how the different effector functions of immune cells and mediators activated in each individual during *Mtb* infection leads to the grouping of TB patients based on endotypes that define their immunological state.

## 3. On the Road to Discover Categorization Systems That Guide TB Host Directed Therapy (HDT)

The current TB therapy regimen has ignored individual patient variation regarding immune responses, susceptibility to infection and the required time of treatment. In general, host immune endotypes might be categorized as immune-deficient or immune-exuberant states of the TB patient [4]. Accordingly, Lange et al. proposed that TB patients’ care may be established according to areas related to the host, such as the immune response and zones associated with the pathogen—for example, the drugs and length of the antibiotic therapy. In effect, the host immune response might be detrimentally suppressed and require a boost, or it can be harmfully exaggerated and need to be suppressed. Such stratified therapies are within reach and could soon become available for the clinical management of TB [4]. 

Immune endotypes might be immune deficient or immune vigorous [4], as the described deficiencies in IL-12, IFN-γ or TNF pathways that lead to the decreased intracellular killing of *Mtb* [18,19] or the exuberant IFN-γ or TNF profiles that were shown to be detrimental since they resulted in pulmonary damage and escape of viable *Mtb* [20,21]. HDT might provide an unexploited approach as complementary anti-TB therapies, either by increasing the ability of the host immune system to eliminate mycobacteria or by limiting collateral tissue damage associated with infection [22,23,24]. Then, as mentioned above, HDT could improve the efficacy and shorten the duration of the actual treatment regimens [13], whereas endotypes would guide the application of personalized immunotherapies [5]. However, to implement endotype-specific HDT, it is necessary to perform a proper characterization of the diverse TB patient populations that would allow us to identify variable targets. Furthermore, new HDT strategies such as the modulation of autophagy as an adjuvant therapy might improve TB patients’ treatment. 

Specific endotypes remain to be defined for most infections, even for those with ostensibly similar clinical phenotypes. In contrast, in asthma, most cancers or chronic obstructive pulmonary disease, biological endotypes have been identified. In those diseases, the defined endotypes display diverse molecular and cellular pathologies that led to an analogous disease phenotype. Then, the treatment of those diseases depends on the specific disturbed molecular pathways and endotype-specific therapies can then be employed [25]. For example, all biological therapies that have been approved in asthma are directed at the T2-high eosinophilic endotype [25]. However, no comparable categorization system is still available to guide TB HDT.

## 4. Empowering Immune Endotypes That Endorse the Clinical Phenotypes of TB Patients during Active *Mtb* Infection

Several studies have contributed to investigate whether diverse immune endotypes could reflect the different clinical phenotypes detected in TB [13]. Accordingly, some years ago, two populations of TB patients were described in relationship to their clinical phenotypes and immune endotypes. Based on their T-cell immune responses to *Mtb* antigens (*Mtb*-Ag), patients were classified as High Responder (HR) or Low Responder (LR) individuals. Briefly, HR TB patients are subjects displaying significant proliferative responses, IFN-γ production and an increased percentage of SLAMF1^+^ CD3^+^ cells after *Mtb*-Ag stimulation; on the other hand, LR TB patients exhibit low proliferative responses, IFN-γ release and amounts of SLAMF1^+^ CD3^+^ cells. Cut-off values to differentiate between the two populations of LR TB and HR TB patients were previously established by Pasquinelli et al. and Amiano 2022 [26,27]. The fulfilment of two of the three mentioned criteria was required to assign a patient to the corresponding group [26,27]. Besides, it was demonstrated that common clinical parameters analyzed in TB patients paralleled immunological criteria studied in HR and LR individuals [7,26]. HR TB patients had significantly higher percentages of total lymphocytes and greater PPD diameters compared with LR. LR individuals displayed more severe pulmonary lesions, a striking weight loss and had been ill longer than HR [7,26]. 

Recently, Di Nardo et al. used unbiased clustering to identify individual TB endotypes with distinctive patterns of gene expression and clinical outcomes [13]. They described a group of individuals with endotype A who displayed increased inflammation and immunity and decreased metabolism and proliferation, whereas another group of patients belonged to endotype B and presented increased metabolism and proliferation. These findings suggest that endotypes are the reflection of the functional immunity of TB patients [13]. The authors hypothesized that based on the augmented predicted treatment failure and disease severity signatures, endotype A patients would display worse clinical outcomes as compared to endotype B patients. Actually, the endotype A patients had slightly increased rates of cavitary disease, higher initial bacterial load, slower times to culture conversion and decreased rates of cure outcomes (Figure 1). Furthermore, by stimulating whole blood with PHA and assaying six different cytokines and chemokines, two TB patient sub-groups were then identified: TB hyperinflammatory/hyperresponsive patients that demonstrated a baseline hyperinflammatory condition similar to that of endotype A but displayed a decreased capacity to up-regulate IFN-γ, TNF, IL-1β, IL-6, CXCL9, and CXCL10, and TB Responsive patients, the immune-responsive group with the capacity to react to stimulation [13] (Figure 1). In contrast, Quiroga et al. reported that the failure of LR TB patients to augment the T-cell expression of the costimulatory molecule ICOS up to the levels displayed by HR TB patients was related to a specific unresponsiveness of LR TB to *Mtb*-Ag. Actually, the authors demonstrated that PHA stimulation of cells from TB patients and healthy donors induced similar augmented levels of ICOS in the three groups of studied individuals, which demonstrated a weak *Mtb*-specific TCR signal in the LR TB patients [28]. 

Additionally, at least three mutually non-exclusive endotypes in active TB have recently been proposed: one caused by IL-12-IFN-γ signaling defects, one characterized by exuberant hyperinflammation and one by immune exhaustion [5]. The authors state that while certain endotypes are exclusive, other endotypes overlap. In fact, defects in the IL-12-IFN-γ axis led to immune deficiency and intersect with the IFN-γ-deficient endotype. Likewise, upon boost antigenic stimulation, immune exhaustion results in both TNF and IFN-γ deficiencies [5]. Finally, regarding the exuberant hyperinflammation, the authors highlight that immunity against mycobacteria involves proper-balanced myeloid and lymphoid immunity, which is lost during the relatively rare manifestation of TB, hemophagocytic lymphohistiocytosis (HLH). The destructive disorder HLH is characterized by clinical and laboratory evidence of extreme inflammation manifested by deficient cytotoxic T-cell and NK cell immunity and immunopathology by excessive TNF, IL-6, and phagocytosis [5].

The comparative analysis of cytokines elaborated in TB across the disease spectrum might provide additional insights into the pathogenesis of the disease. Applying the WHO criteria for the stratification of patients according to the severity of the disease (EPTB1 = disseminated and/or severe disease; PTB = disease localized to lung parenchyma, EPTB2 = disease localized to peripheral sites without lung involvement), it was demonstrated that *Mycobacterial* antigens induced an IFN-γ/IL-10 ratio in direct correlation with the disease severity [32]. In fact, while peripheral blood mononuclear cells from mild TB patients produced Th1 cytokines upon *Mtb*-Ag, both Th1 and Th2 cytokines and elevated production of TGF-β were measured in more advanced stages of the disease [33]. Therefore, the protective cellular immune response would be progressively lost as the disease progresses. 

TB patients showed important differences regarding X-ray radiography [33] and leukocyte count [7]. An immunophenotypic characterization of unilateral and bilateral active TB (according to the extent of lung lesions) performed in a population of Brazilian patients showed that the magnitude of lung lesions could be associated with a fine-tuning between immunological responses during untreated *Mtb* infection [34]. Interestingly, significantly augmented levels of both IL-17^+^CD4^+^T-cells and IL-17^+^CD8^+^T-cells were detected in the bilateral group as compared to the unilateral group, suggesting a potential deleterious role of these subsets during active pulmonary *Mtb* infection [34]. Other studies have suggested the involvement of Th17 in TB pathology. Actually, Jurado et al. demonstrated an increased Th17 response in TB patients, showing that IL-17 was mainly secreted by IFN-γ^+^IL-17^+^ CD4^+^ T lymphocytes. Interestingly, the highest fraction of IFN-γ^+^IL-17^+^ CD4^+^ T cells was detected in LR TB patients. Moreover, the authors demonstrated a positive association between two clinical parameters (time of disease evolution and pulmonary lesions) and the proportion of CD4^+^IFN-γ^+^IL-17^+^ lymphocytes expanded in response to *Mtb*-Ag. Actually, they described that LR TB patients showed higher ratios of Th1/Th17 cells in pleural fluids and peripheral blood as compared with HR patients. Then, these results provide evidence that the proportion of IFN-γ^+^IL-17^+^ CD4^+^ T lymphocytes in patients with active TB might be directly associated with the severity of the disease [7] (Figure 1). 

Recently, Domaszewska et al. found that IFN responses were not equally distributed among TB patients [35]. Rather, in a subgroup of TB patients, IFN responses were the dominant immune responses and defined the IFN^+^ group of TB patients. In contrast, IFN responses were less pronounced in the group defined as IFN^−^. The distinct population of TB patients who did not develop IFN responses detectable by gen set analysis presented less severe lung pathology. The authors explained that two types of IFN signaling are considered crucial for the outcome of TB: (i) the IFN I signaling pathway is thought to be mostly detrimental, and (ii) the IFN II pathway is considered to play a major role in protection. However, their analyses revealed that the majority of TB cases had shared IFN I and IFN II responses, indicating that both detrimental and beneficial mechanisms coexist in active TB disease. Consistently, lung pathology was found to be more predominant in the IFN^+^ TB patient group as compared to the IFN^−^ TB patient group. Domaszewska et al. concluded that the balance between both types of IFN probably determines the outcome of the infection. Therefore, HDT will most likely diverge in these two endotypes of active TB, with the IFN^+^ group likely benefiting from IFN dampening and the IFN^−^ group probably from IFN-promoting therapy [35]. 

## 5. Differential Clinical Outcomes among TB Endotypes: Activation of Myeloid and Lymphocytic Immune Cells from Patients to Fight *Mtb*

Several signaling proteins are known to modulate the level and pattern of cytokines produced by T cells upon *Mtb*-Ag stimulation [36,37]. Thus, the study of these molecules could contribute to a better understanding of the categorization of the different immuno-endotypes in TB. Actually, SLAMF1 and ICOS activation increased Th1 lymphocytes against *Mtb*-Ags [26,28], whereas PD-1, SAP and CD31 inhibited this population [38,39]. Pasquinelli et al. demonstrated that *Mtb*-Ag-induced IFN-γ was impaired in cells from TB patients displaying SAP, while cells expressing SLAMF1 alone secrete high IFN-γ amounts against the pathogen [26]. Furthermore, SLAMF1 ligation significantly enhanced IFN-γ production in HR TB patients and slightly augmented IFN-γ secretion in LR patients. Remarkably, SAP expression in LR TB patients displayed a striking down-regulation after the ligation of SLAMF1 [26]. Thus, those findings indicate that SAP inhibits IFN-γ production during mycobacterial infection, whereas SLAMF1 activation improves cell-mediated immunity to *Mtb* infection. In addition, Quiroga et al. reported that patients with robust Th1 cytokine production in response to *Mtb* (HR patients) express high levels of ICOS in contrast to low levels of both IFN-γ and ICOS in LR TB patients [28]. Then, efficiently Ag-activated T cells would up-regulate ICOS, and therefore, ICOS ligation would augment IFN-γ secretion, while *Mtb* weak-responsive T cells would fail to secrete IFN-γ against the pathogen, leading to reduced ICOS levels. Quiroga et al.’s results identified for the first time that ICOS activation was related to Th1 cell responses in human intracellular infection and might amplify those responses [28]. In addition, Jurado et al. demonstrated a direct association between PD-1 and IFN-γ at the site of infection in TB patients, indicating that T cells from HR patients were systemically activated by the specific Ag, inducing IFN-γ and generating a Th1-like microenvironment that in turn led to heighten PD-1 expression. In contrast, systemic T-cells from unresponsive TB patients (LR) were weakly activated by *Mtb*-Ag and produced IL-10 but very low amounts of IFN-γ, leading to a predominant Th2-like environment that impaired PD-1 increase on their T-cells. Even at the site of infection, LR TB patients produced diminished levels of IFN-γ [38]. Importantly, simultaneous PD-1 blockage and co-stimulation through SLAMF1 **increased** the IFN-γ secreted by cells from pleural fluids and peripheral blood in LR TB patients (subjects with the non-responder endotype) up to the levels produced by HR individuals (individuals with the responder endotype), demonstrating the relevance of the balance between costimulatory molecules and checkpoint proteins in determining the anti-TB host response. Furthermore, similar results were found by blocking PD-1 together with ICOS activation [38]. Then, the use of blocking and agonistic mAb might have important implications for potential therapies for chronic infections such as TB, as it was previously reported for other diseases [40]. 

The role of B-cells and antibodies during TB infection have been ignored for a long time. The evidence for the contribution of B-cells and antibodies to the clearance of *Mtb* varies significantly, in part as a result of genetic or environmental differences in the populations studied, which are particularly present in the diverse populations studied in the context of TB. Thus, more global evaluations of the effector responses, including humoral responses, are required [41]. Given that *Mtb* is extracellular during its reinfection phase and during expectoration, it is susceptible to various mechanisms of antibody-mediated immunity, such as mycobacterial neutralization, antibody-dependent cellular phagocytosis, complement activation, and antibody-dependent cell-mediated cytotoxicity (ADCC), among others [41]. It has been proposed that whereas *Mtb*-specific antibodies might be involved in bacterial clearance and immune control, an enhanced activation of antibody responses in human TB could reflect an impaired cellular immunity. In this way, during the chronic phase of TB, infection-elevated antibody responses might indicate exacerbated disease [42]. Accordingly, Ashenafi et al. showed that the high levels of *Mtb*-specific IgG-secreting cells in the peripheral circulation of TB patients were associated with reduced *Mtb*-specific IFN-γ production and severe disease [43]. Although B-cells and antibodies may not be entirely sufficient by themselves, they definitely might contribute to the complete effector response, providing new targets for future tools to fight *Mtb*. Moreover, Zhang et al. reported that TB patients had significantly higher frequencies of CD19^+^CD1d^+^CD5^+^ B cells, with stronger suppressive activity than such cells from healthy donors. In addition, the levels of those regulatory B cells were directly associated with disease severity. Furthermore, the authors demonstrated that primary CD19^+^ B cells from TB patients significantly inhibited Th17 but not Th1 cell activation. However, more information about the relationships between B- and T-cell subsets and their relative contributions to the modulation of immune responses during *Mtb* infection is required [29].

Neutrophils are the cells that arrive earlier at the site of infection and are mainly infected with *Mtb* in TB patients’ lungs [44]. The process of autophagy is essential for many functions of neutrophils [45]. Recently, Pellegrini et al. demonstrated that *Mtb*-stimulation significantly induced SLAMF1 expression in human neutrophils. Interestingly, the authors detected that *Mtb*-Ag-stimulation induced superior levels of SLAMF1^+^ in neutrophils from HD as compared to neutrophils from TB patients. Furthermore, Pellegrini et al. demonstrated that *Mtb*-Ag induced autophagy in neutrophils from TB patients, and SLAMF1 ligation further augmented the process in these cells. Moreover, greater autophagy levels were measured in neutrophils from HR TB patients (those individuals associated with the responder endotype) as compared to LR TB patients [46]. Taking into account that autophagy protects against excessive inflammation during *Mtb* infection [46], the reduced autophagy detected in LR TB patients’ neutrophils might be associated with the common detrimental inflammatory responses occurring during active disease. Therefore, increased autophagy in neutrophils from patients with the responder endotype might benefit degranulation, reactive oxygen species (ROS) production, and the release of NETs, all collaborating on the outcome of the disease. 

One of the main goals of HDT strategies is to augment the patient’s ‘immunity. The induction of phagosome acidification and autophagy in *Mtb*-infected macrophages are important effector mechanisms of host resistance to infection, which might be attractive targets for HDT strategies. Tyrosine kinase inhibitors are one class of drugs that promote the macrophage phagocytic response in TB [47]. Inhibitors of mechanistic target of rapamycin (mTOR) are a second group of agents thought to promote the macrophage-mediated control of *Mtb*, in this case, through the induction of autophagy [48]. However, none of these findings reported in macrophages consider the immune endotypes in TB. Rovetta et al. reported a differential pattern of autophagy induction by IFN-γ in monocytes from TB patients with different immune endotypes [30]. *Mtb*-Ag stimulation of monocytes induced the highest levels of autophagy in healthy donors and the lowest levels in the LR TB patients, in direct association with the amounts of IFN-γ secreted by each individual. Interestingly, when *Mtb*-Ag was added with exogenous recombinant IFN-γ to cells from LR patients, LC3-II levels increased up to the levels detected in Ag-stimulated cells from HR patients. Therefore, autophagy in *Mtb*-Ag-stimulated monocytes is influenced and regulated by Th1 responses against the pathogen, denoting the importance of host-cell-mediated immunity in TB [30]. Furthermore, it was reported that IL-17A augmented autophagy in *Mtb* H37Rv-infected monocytes from HR TB patients through a mechanism that activated the MAPK1/ERK2-MAPK3/ERK1 pathway. In contrast, during infection of monocytes from LR patients, IL-17A had no effect on the autophagic response. However, the addition of IFN-γ to *Mtb* H37Rv-infected monocytes increased autophagy both in HR and LR TB patients through the activation of the MAPK14/p38α pathway. Therefore, according to the different immune endotypes in TB patients, IL-17A was unable to augment autophagy in non-responder patients because of a defect in the MAPK1/3 signaling pathway, whereas both IFN-γ and IL-17A increased autophagy levels in patients with a strong immunity to *Mtb*, promoting mycobacterial killing [31]. Interestingly, it has been demonstrated that clinical isolates of *Mtb* differed in their ability to induce autophagy and that patients infected by strains displaying poor autophagy-inducing ability exhibited more severe disease. In the mentioned study, the radiographic extent of active TB was characterized as “minimal”, “moderately advanced”, or “far advanced” [49] and the effect of different *Mtb* isolates from TB patients on autophagy induction was studied by infecting human THP-1 macrophages. Then, Li et al. showed that the capacity of different *Mtb* isolates to induce autophagy correlated with clinical outcomes in TB patients [50]. These findings demonstrate the influence of bacterial genotype on the differential host response. 

Although individualized therapies are considered impractical, it will be especially important to identify clinical correlates of specific endotypes to select appropriate immunotherapies for particular patients. For example, circulating CD16^+^ monocytes and monocyte-like myeloid-derived suppressor cells (M-MDSC) have been associated with clinical parameters of TB disease severity [30,31]. Moreover, LR TB patients, individuals with a weakened immune response [7,26] showed higher proportions of intermediate and non-classical monocytes in their peripheral blood as compared to HR TB patients (subjects with a strong immunity). Thus, the proportion of circulating CD14^++^CD16^+^ and CD14^+^CD16^++^ monocytes in patients with active TB might reflect the immunological and clinical severity of the disease [27]. Recently, Amiano et al. described a characteristic expansion profile of M-MDSC and polymorphonuclear-like (PMN)-MDSC cells according to the immune endotypes of TB patients [27]. Actually, HR TB patients displayed augmented PMN-MDSC, whereas LR TB individuals showed the highest levels of circulating M-MDSC. However, an inverse correlation between circulating M-MDSC and IFN-γ index was detected only in HR TB patients, suggesting that M-MDSC from HR TB would present more suppressive effects than LR TB’s M-MDSC. Although pharmacological therapy was reported to reduce the accumulation of myeloid cell populations [51,52,53], Amiano et al. observed a restoration in CD14^++^CD16^+^ and CD14^+^CD16^++^ monocytes and M-MDSC after anti-TB treatment. Then, determining the levels of CD14^++^CD16^+^ and CD14^+^CD16^++^ monocytes and M-MDSC and the information regarding the immunological status of TB patients might allow the differentiation of responder and non-responder TB patient endotypes [27]. 

## 6. Extrapulmonary TB: A Special Case of Active TB

Recently, Fiske et al. reported an increase in Vitamin D receptors (VDRs) in macrophages from individuals with previous extrapulmonary TB (EPTB). EPTB concerns organs other than the lungs, for example the lymph nodes, pleura, bones, joints, meninges, the central nervous system, and gastrointestinal or genitourinary areas. EPTB disease accounts for about 20–50% of reported TB cases [54] and was shown to be more frequent in patients with underlying immune defects, such as HIV positive people [55], young children or other immunocompromised individuals. Diagnosis of EPTB is often more difficult than pulmonary TB because patients frequently present negative sputum-based tests. However, the immune landscape of segregated endotypes of EPTB has been insufficiently investigated as compared with pulmonary active TB, comprising a topic of interest for researchers and clinicians. In this regard, Blankey et al. analyzed the whole blood transcriptional response of EPTB patients compared to pulmonary TB patients and showed that this response is similar across different manifestations of the disease, although highly variable, and is mainly linked to the presence or absence of symptoms [56]. Additionally, McCaffrey and colleagues aimed to characterize the deep architecture and composition of human granulomas by using multiplexed ion beam imaging by time of flight (MIBI-TOF) [57]. When comparing granulomas from the lungs and extrapulmonary sites, the authors found that the majority of cell types were present in similar proportions. However, increased proportions of mast cells, CD68^+^ macrophages and multinucleated giant cells were detected in pulmonary tissues. Similarly, the presence of CD14^+^ CD16^+^ intermediate monocytes was almost exclusive to extrapulmonary tissues. Furthermore, it was previously reported that individuals that had EPTB displayed decreased cytokine production and lower CD4^+^ lymphocyte counts compared to persons with previous pulmonary TB, LTBI and uninfected contacts [58,59,60]. Moreover, Fiske et al. had also reported that people with previous EPTB had increased T-cell activation and augmented regulatory T-cell frequency as compared to the other patient groups [60]. The limitation on the heterogeneity observed and the number of EPTB cases analyzed in the literature so far prevents the possibility of reliably classifying patients with EPTB according to their immune responses or even associating them with endotypes already described in patients with pulmonary TB.

## 7. Immunotherapy Research in Tuberculosis

Immunotherapy directed at *Mtb* infection focuses on enhanced protective immunity, suppressed adverse immune responses and inflammatory damage [61]. Therefore, the development of TB-specific immunotherapies might contribute the modulating of the host immune response against the pathogen, leading to new combined strategies that allow us to shorten the current chemotherapy used for TB. The practice and potential of immunotherapy in a bacterial infectious disease such as TB includes strategies to augment immunity and strategies to ameliorate immunopathology [47]. Moreover, HDTs that modulate host–pathogen interactions offer an innovative strategy to combat *Mtb* infections. In fact, the induction of autophagy, employment of mTORC1 inhibitors, modulation of cytokine responses, enhancing T-cell mediated host responses, regulation of host epigenetics, among others approaches, have been proposed as potentially useful HDT schemes [47]. Accordingly, several clinical trials have been developed employingimmune-active substances, cellular therapies, chemical agents and vaccines [61]. In fact, key approaches to HDT were assayed with cytokines such as IL-2 and GM-CSF [62,63] or anti-tumor necrosis factor; with macrophage-targeting strategies (tyrosine kinase inhibitors and mechanistic target of rapamycin (mTOR) inhibitors; modulation of autophagy) [47]; Vitamin D [64]; and anti-inflammatory drugs (for example, statins [65] and cyclooxygenase 2 inhibitors [66]) and therapeutic vaccines [61]. In summary, immunotherapy directed to the host provides a unique potential treatment strategy to combat *Mtb* infection as an adjuvant of classical anti-TB chemotherapy. However, HDT is challenged by complex interactions between drugs, the host and the pathogen. Furthermore, as mentioned above, to put into practice endotype-specific HDT, a proper characterization of the diverse TB patient populations that would allow us to identify variable targets is required. Moreover, at present, there is a lack of immunodiagnostic methods that allow a follow-up of TB progression that would permit us to provide a correct application of an immunotherapy [61]. Although many crucial mechanisms might be exploited as HDT, translation towards clinically effective treatment strategies in active TB is still complicated [67]. 

## 8. Concluding Remarks and Future Perspectives

The differential activation of immune cells in each TB patient will depend on the subject’s cellular metabolism, genetic background, epigenetic patterns and gene transcriptional regulation. Moreover, the genotype of *Mtb*, the bacillary load, the patient’s lung damage and different environmental factors will also modulate the diverse immune cells’ activation in the individual. Together, all the mentioned factors will define the patient’s immune endotype. Diverse TB immune endotypes will be then reflected on the clinical phenotype, delineating the severity of the disease in the *Mtb*-infected individual. Clearly, contrary to works employing animal models, those including human beings denote a huge heterogeneity that makes it unfeasible to give the same treatment to everyone. Thus, achieving the identification of each patient’s immune endotype would be key to prescribing specific successful therapy. However, to date, the management of TB patients does not consider the variability of the host or the pathogenicity of *Mtb*. Therefore, the introduction of personalized medicine concepts may allow for individual selection of the best therapy regimens [4]. However, much progress still needs to be made. Accordingly, it is necessary to find precise and accessible markers that will allow for the identification of immune endotypes in TB patients in health centers. In particular, it is crucial to determine the immune endotypes of patients with impaired cell mediated immunity against *Mtb* and the way to improve their defense mechanisms against the pathogen. In cancer medicine, enormous collaborative efforts led to identifying distinct endotypes, achieving targeted endotype-specific therapies that improved clinical outcomes. Thus, the identification of different TB immune endotypes, together with their host-related, environmental and pathogen-associated factors, would allow us to recognize TB immune endotype-specific therapies that will improve clinical outcomes.

## Figures and Tables

**Figure 1 ijms-24-05033-f001:**
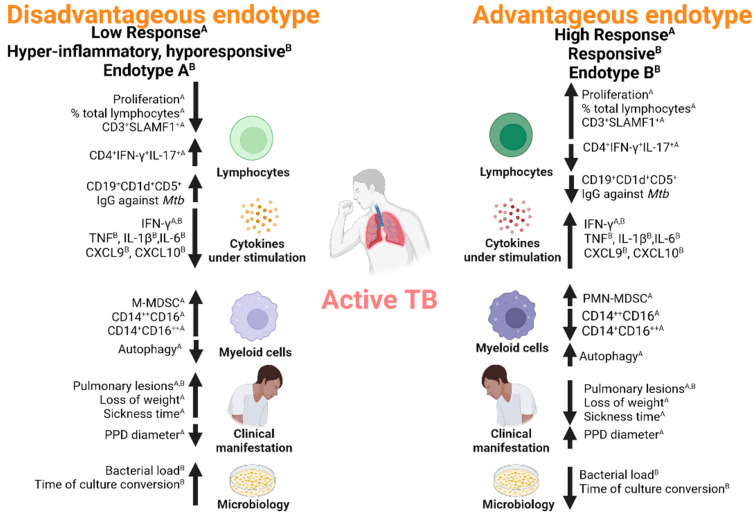
Characterization of immune endotypes in patients with active TB. The figure shows schematic categorization examples of immune endotypes in TB. The Disadvantageous and Advantageous endotypes are displayed as: (i) Low Responder/High Responder types defined according to the criteria indicated as (A) [26]; (ii) Hyporesponsive/Hyper-inflammatory, hyporesponsive; and (iii) Endotype A/Endotype B defined according to the criteria indicated as (B) [13]. The Disadvantageous endotype includes TB patients with weak or no T-cell responses to *Mtb*-Ag, more severe disease [7,26], high levels of CD19^+^CD1d^+^CD5^+^ regulatory B-cells [29] and decreased levels of cytokines released [7,13,26]. Moreover, M-MDSC and intermediate and non-classical monocyte populations are augmented in the Disadvantageous endotype [27]. Furthermore, autophagy is diminished in this endotype [30,31]. The Advantageous endotype corresponds to TB patients that display strong T cell immunity to *Mtb*-Ag, milder manifestations of the disease [7,26] and increased levels of cytokines released [7,13,26]. Additionally, the PMN-MDSC are augmented in the Advantageous endotype, the intermediate and non-classical monocyte populations are diminished [27] and robust autophagic responses are observed [30,31]. Figure created with BioRender.com.

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
