# Peer review of "Beyond the Clinic: The Activation of Diverse Cellular and Humoral Factors Shapes the Immunological Status of Patients with Active Tuberculosis"

_ijms, 2023, doi:10.3390/ijms24055033_

Round 1

Reviewer 1 Report

This manuscript can be accepted after a few language check and revisons. 

Author Response

Response to Reviewer 1 Comments:

This manuscript can be accepted after a few language check and revisions. 

We thank the Reviewer for the decision about our manuscript.  We have carefully checked the grammar of the work and we have corrected all the language and spelling mistakes. 

Reviewer 2 Report

The idea of the manuscript is excellent, but the authors on several occasions cited their works through a descriptive presentation without citing specific results.

It would be good if the authors would slightly reorganize their manuscript in the direction of presenting concrete results and discussing them.

Author Response

Response to Reviewer 2 Comments:

The idea of the manuscript is excellent, but the authors on several occasions cited their works through a descriptive presentation without citing specific results.  It would be good if the authors would slightly reorganize their manuscript in the direction of presenting concrete results and discussing them. We appreciate very much the opinion of the Reviewer about our work.  Following her/his advice we have now included specific results about our previous works, in order to clarify and complete the original descriptive presentation.   Then, we have reorganized the manuscript including concrete results and also discussing them.  The changes are highlighted in bold in the revised manuscript. 

Reviewer 3 Report

This manuscript gave the overview of the studies describing the functions of immune cells and mediators in tuberculosis, which may lead to the classification of patients into immune endotypes, which define their immunological state and may be useful in personalized medicine strategies.

Overall, this manuscript is well written, provides much useful information and it brings some novelty in the research field. However, there are several concerns that need to be clarified.

In order to be comprehensive, this review article should contain one new part before the concluding remarks on the emerging immunotherapy strategies. Authors stated in the text: “Although individualized therapies are considered impractical, it will be especially important to identify clinical correlates of specific endotypes to select appropriate immunotherapies for particular patients”. Therefore, in order to give practical significance to this manuscript, the current state of immunotherapy research in the field of tuberculosis should be added.

Some other concerns include:

Please revise the epidemiological data. According to the WHO, tuberculosis is the 13th leading cause of death and the second leading infectious killer after COVID-19 (https://www.who.int/news-room/fact-sheets/detail/tuberculosis). It should be corrected in the text (lines 18 and 34).

There are some technical issues in the manuscript, such as full stops at the end of all titles, no space before square bracket with reference numbers, improper font (in Funding), some grammatical and linguistic errors, etc. Paragraphs at lines 140 and 141 should be merged.

The list of abbreviations should be added, or at least some terms should be defined in text (like PHA, mAb and several other).

Author Response

Response to Reviewer 3 Comments:

Overall, this manuscript is well written, provides much useful information and it brings some novelty in the research field.

 We thank the Reviewer for her/his opinion about our manuscript.

However, there are several concerns that need to be clarified.

1-In order to be comprehensive, this review article should contain one new part before the concluding remarks on the emerging immunotherapy strategies. Authors stated in the text: “Although individualized therapies are considered impractical, it will be especially important to identify clinical correlates of specific endotypes to select appropriate immunotherapies for particular patients”. Therefore, in order to give practical significance to this manuscript, the current state of immunotherapy research in the field of tuberculosis should be added. The Reviewer´s suggestion is certainly a very good idea. And although the immunotherapy in TB is not precisely the scope of the manuscript, we agree with the Reviewer that a paragraph on this issue will improve the paper and will give it practical significance.  Therefore, we have now included a new part before the concluding remarks to discuss the state of immunotherapy research in the field of tuberculosisWe entitled it "Immunotherapy research in tuberculosis" and it is highlighted in bold in the revised manuscript.  We tried to summarized  the state of the art about  immunotherapy research in human tuberculosis, including the most important advances.  However, if the Reviewer considers that the section should be even more complete, we can include more information about this point. 

2. Some other concerns include:

2.1.Please revise the epidemiological data. According to the WHO, tuberculosis is the 13th leading cause of death and the second leading infectious killer after COVID-19 (https://www.who.int/news-room/fact-sheets/detail/tuberculosis). It should be corrected in the text (lines 18 and 34). We apologize for the mistake.  We have corrected the data according to the Reviewer´s comment in both lines.  The changes are highlighted in bold in the revised manuscript.

2.2. There are some technical issues in the manuscript, such as full stops at the end of all titles, no space before square bracket with reference numbers, improper font (in Funding), some grammatical and linguistic errors, etc. Paragraphs at lines 140 and 141 should be merged. We apologize again for the mistakes mentioned by the Reviewer.  We have corrected full stops at the end of all titles, we have added spaces before square bracket with reference numbers,  we have corrected the font in Funding Section and we have revised all the grammatical and linguistic errors.

2.3. The list of abbreviations should be added, or at least some terms should be defined in text (like PHA, mAb and several other). We have added a list containing the abbreviations mentioned in the manuscript before the References Section. They are highlighted in bold in the revised manuscript.

Round 2

Reviewer 3 Report

The manuscript is improved after revisions and I support it to be published in this form.